# TiO_2_ Thickness-Dependent Charge Transfer in an Ordered Ag/TiO_2_/Ni Nanopillar Arrays Based on Surface-Enhanced Raman Scattering

**DOI:** 10.3390/ma15103716

**Published:** 2022-05-22

**Authors:** Cai Wang, Xufeng Guo, Qun Fu

**Affiliations:** Institute of Nanochemistry and Nanobiology, School of Environmental and Chemical Engineering, Shanghai University, Shanghai 200444, China; thre3@shu.edu.cn

**Keywords:** TiO_2_-thickness dependence, charge transfer, SPR, interface, SERS

## Abstract

In this study, an ordered Ag/TiO_2_/Ni nanopillar arrays hybrid substrate was designed, and the charge transfer (CT) process at the metal–semiconductor and substrate–molecule interface was investigated based on the surface-enhanced Raman scattering (SERS) spectra of 4-Aminothiophenol (PATP) absorbed on the composite system. The surface plasmon resonance (SPR) absorption of Ag changes due to the regulation of TiO_2_ thickness, which leads to different degrees of CT enhancement in the system. The CT degree of SERS spectra obtained at different excitation wavelengths was calculated to study the contribution of CT enhancement to SERS, and a TiO_2_ thickness-dependent CT enhancement mechanism was proposed. Furthermore, Ag/TiO_2_/Ni nanopillar arrays possessed favorable detection ability and uniformity, which has potential as a SERS-active substrate.

## 1. Introduction

Surface-enhanced Raman scattering (SERS) has become one of the most powerful tools to characterize interfacial charge transfer (CT), trace detection and catalytic monitoring because of its unique fingerprint spectroscopy characteristics and highly sensitive detection capabilities [1,2,3]. Since SERS was first observed in 1974 [4] and developed so far, many efforts have been devoted to improving SERS sensitivity, which is inseparable from the exploration of enhancement mechanisms. It is commonly accepted that electromagnetic enhancement (EM) and chemical enhancement (CM) contribute to the satisfactory enhancement of SERS. EM is mainly attributed to the collective oscillation of free electrons on the plasmonic metal surface under the excitation of incident light to generate surface plasmon resonance (SPR), which greatly enhances the strength of the electromagnetic field on the metal surface [5,6]. Previous studies proved that SPR depends to a large extent on parameters such as the morphology, size, thickness and spacing of plasmonic materials, while the relationship between parameters regulation of metal and SERS performance has also been widely discussed [7,8]. CM is related to the CT process of molecular chemisorption on the surface of the substrate material, which increases the polarizability of the molecules and selectively enhances the non-totally symmetric vibrational modes in SERS spectra [9,10,11]. The enhancement of the interfacial CT transition changes the shape of the Raman spectrum, making it particularly important in studying substrate-molecular interfacial interactions.

Metal oxide semiconductors (TiO_2_, ZnO and Cu_2_O) are widely used as SERS substrate materials owing to their excellent chemical stability, good biocompatibility and selectivity [12,13,14]. Excitingly, a few semiconductors were found to exhibit high SERS enhancement comparable to metals, which has attracted extensive attention from researchers on CT mechanisms [12,15]. Recently, the influence of semiconductor parameters on CT mechanism and SERS has been widely reported [16,17]. Studies based on the size of nanoparticles indicated that SERS activity of adsorbents on TiO_2_ and ZnO exhibited a strong size-dependent CT effect, and the enhanced CT process of Herzberg–Teller coupling vibration was responsible for SERS [18,19]. In addition, the relationship between the thickness of the layered structure semiconductor and SERS enhancement was also revealed. SERS intensity of R6G adsorbed on MoS_2_ decreases significantly with the increase in layers, which is attributed to the different charge transfer degrees between molecules and substrates [20]. Although semiconductor parameter-dependent CT mechanisms were demonstrated, the achievable SERS enhancement of semiconductor oxides is quite limited, which greatly hinders the practical detection and application of semiconductor substrates.

The hybridization of semiconductors with plasmonic metals allows the desirable SERS performance because of the combination of surface plasmon resonance and CT enhancement, which displays distinct advantages compared to single material [21]. As we all know, a strong electric field forms near the surface of the hybrid nanostructure under SPR excitation. The local electric field strength and light absorption are tremendously affected by the properties of the semiconductor dielectric layer, which enables the hybrid system to exhibit new properties and synergistic enhancement effects [22,23]. Additionally, the degree of induced excitation CT process at the metal–semiconductor interface depends on the parameter and dielectric properties of the semiconductor. For example, the UV-vis absorption properties and the chemical stability of Au@Cu_2_O core−shell nanoparticles are found to be tunable with Cu_2_O thickness [24]. Zhao et al. [25] found that the CT degree of the Ag/PATP/ZnO system depends on the size of ZnO nanoparticles, and the possible CT resonance mechanism was responsible for the size-dependent SERS behavior. Studies based on the semiconductor thickness and CT enhancement discovered that the hot electrons transfer efficiency at metal–semiconductor interface is significantly influenced by the regulation of semiconductor thickness, which plays a decisive role in CT contribution to SERS [17,26]. In a 4-MBA/Au/Cu_2_S composite system, the conspicuous Raman enhancement is attributed to the modulation of the Cu_2_S shell thickness, which changes the SPR resonance frequency and the distribution of carriers [27]. The study of SPR and interfacial CT process provides vital information on the interfacial interaction of substrate materials and also provides a theoretical basis for the optimal design of excellent SERS substrates. At present, the effect of the semiconductor thickness on the SERS and photocatalytic performance in hybrid systems has been reported [24,28,29,30]. However, the SPR and CT mechanism, as well as the CT process at the substrate–molecule interface related to semiconductor thickness changes, are rarely studied in the composite system. In addition, for composite systems containing noble metal nanoparticles, high homogeneity and high density of “hot spots” are essential for excellent SERS substrates.

Herein, Ag/TiO_2_/Ni nanopillar arrays were fabricated, and the effect of SPR on the CT mechanism as well as the CT process at the substrate-molecular interface was studied by modulating TiO_2_ thickness. UV-vis and XPS were conducted to reveal the CT process on Ag-TiO_2_ interface, which demonstrated that the electron’s density and SPR absorption are influenced by TiO_2_ Thickness. The CT process between substrates to PATP molecules is confirmed by XPS spectra, and the CT contribution to SERS under different laser wavelengths exhibits a strong TiO_2_-thickness dependence. Meanwhile, Ag/TiO_2_/Ni nanopillar arrays were proved to possess highly sensitive and uniform, which have the potential to be excellent SERS-active substrates. This work provides a new perspective and a simple approach for studying the effect of semiconductors in composite SERS substrates and offers a reference for analyzing the interaction of semiconductors and metals.

## 2. Materials and Methods

### 2.1. Materials

Sulfuric acid (H_2_SO_4_, 98%), Hydrogen peroxide solution (H_2_O_2_, 30%), Isopropyl alcohol (C_3_H_8_O, 99.9%), absolute ethyl alcohol (C_2_H_5_OH, 99.9%), 3-APTES ((3-aminopropyl)triethoxysilane, 99.9%), gold powder (Au, 99.99%), Nickel sulfate hexahydrate (Ni_2_SO_4_·6H_2_O, 99.9%), Nickel chloride hexahydrate (NiCl_2_·6H_2_O, 99.9%), Boric acid (H_3_BO_3_, 99.5%), silver powder (Ag, 99.99%) and 4-aminothiophenol (PATP, 99.8%) were used. The deionized (DI) water with a resistivity of 18.2 MΩ·cm produced using a Milli-Q SP ultrapure-water purification system from Nihon Millipore Ltd., Tokyo, Japan, was used for cleaning and solvent. Silicon mode was purchased from Fachgebiet Angewandte Nanophysik of Technische Universität Ilmenau, Ilmenau, Germany.

### 2.2. Preparation of Ni Nanopillar Arrays

The Ni nanopillar arrays were fabricated utilizing a commercial silicon mold with nanopit arrays, and the period of the nanopit arrays was 400 nm. Firstly, the silicon was cleaned by a series of pretreatment and modified in ethanol solution with 1% 3-APTES (volume ratio) at 60 °C for 1 h [22]. Then the modified silicon was cleaned in isopropanol and dried. After that, a gold layer of 20 nm thickness was evaporated on top of the silicone mold by Physical Vapor Deposition (PVD) as a conductive layer of electrochemical deposition. (The thickness of Au was measured by the crystal thickness monitor of the PVD equipment.) Ni was then electrodeposited on the silicon mold in the electroplating solution composed of Ni_2_SO_4_·6H_2_O (0.38 M), NiCl_2_·6H_2_O (0.12 M) and H_3_BO_3_ (0.5 M) at current densities of −2 mA cm^−2^, −4 mA h cm^−2^ and −10 mA cm^−2^, and −150 mA h cm^−2^, respectively. Finally, Ni nanopillar arrays were obtained by peeling off Ni film from the surface of silicon.

### 2.3. Preparation of Ag/TiO_2_/Ni Nanopillar Arrays

TiO_2_ with different thicknesses (10, 20, 30, 40, 50 and 60 nm) were deposited on the prepared Ni nanopillar arrays by atomic layer deposition (ALD) system. In the G2 Savannah ALD system, the highly purified precursor Tetrakis(dimethylamino)titanium (Ti(NMe_2_)_4_) and high purity water were used as Ti and O sources, respectively. In the ALD process, the reaction chamber temperature was stabilized at 250 °C. The growth rate was 0.45 Å per cycle, and the numbers of deposition cycles were 222, 445, 667, 889, 1111 and 1333. After deposition, TiO_2_/Ni nanopillar arrays with different TiO_2_ thicknesses were obtained. Ag nanofilm with an evaporation thickness of 15 nm was prepared in a PVD system; the voltage V = 0.45 V, the current I = 5.00 A and the evaporation rate was approximately 0.05 nm/s under the condition of 5 × 10^−4^ Pa chamber vacuum. After evaporation, Ag/TiO_2_/Ni nanopillar arrays with different TiO_2_ thicknesses were obtained.

### 2.4. SERS Measurements

SERS measurements were conducted on an inVia Qontor/Renishaw system with a laser excitation wavelength of 532 and 633 nm. All measurements were performed by a 50× objective lens with a laser power at ~5 mW, and the Raman spectra were obtained at an exposure time of 1 s. The prepared Ag/TiO_2_/Ni were immersed in PATP/ethanol solution (10^−5^–10^−11^ M) for 1 h and then washed with deionized water 3 times to eliminate the excess molecules and dried at room temperature. In addition, at least 6 samples were prepared for each thickness and solution concentration to ensure the representation of results. The baseline of Raman spectra was automatically corrected by the inVia Qontor/Renishaw system to exclude background interference.

### 2.5. Materials Characterization

The morphology characterization and elemental analysis of Ni, TiO_2_/Ni and Ag/TiO_2_/Ni were performed on Field Emission Scanning Electron Microscopy (FE-SEM, Nova NanoSEM 450 (FEI Company, Hillsboro, America)). The UV-vis absorption of the samples was measured on UV-vis spectrophotometry (UV-vis, UV-3600 (Shimadzu company, Kyoto, Japan)) in the wavenumber range of 200–800 nm. XPS studies were performed on X-ray photoelectron spectroscopy (XPS, Thermo Scientific K-Alpha (Thermo Fisher Scientific, Waltham, America))to analyze the elemental composition and chemical states of the samples.

## 3. Results and Discussion

### 3.1. Morphology and Structure Characterization of Ag/TiO_2_/Ni Nanopillar Arrays

ALD can fabricate pinhole-free semiconductor thin films with controllable thickness at the single-molecule level, which shows a unique advantage in studying the CT process [31]. Therefore, ALD is chosen to control the thickness of the semiconductor in the system precisely. The whole fabrication process of Ag/TiO_2_/Ni nanopillar arrays and SERS measurement of PATP molecules are shown in Figure 1. The Ni nanopillar arrays were fabricated by a long-range ordered imprinted silicon template. The 10–60 nm TiO_2_ films were precisely deposited on the surface of Ni Nanopillar arrays in the ALD system. Ag/TiO_2_/Ni nanopillar arrays were obtained by depositing 15 nm Ag nanofilm on TiO_2_/Ni nanopillar arrays in the PVD system, and the SERS performance of PATP molecules absorbed on Ag/TiO_2_/Ni nanopillar arrays was measured under 532 and 633 nm excitation laser wavelength.

The SEM images of Ni, TiO_2_-40/Ni and Ag-15/TiO_2_-40/Ni nanopillar arrays are shown in Figure 2a–c. It can be seen that the fabricated Ni nanopillar array is highly uniform and arranged with a period of 400 nm in a long range. Compared to the Ni nanopillar array, no significant changes are observed; this could be attributed to the homogeneity of the TiO_2_ layer. After the evaporation of 15 nm Ag, Ag nanoparticles are uniformly distributed on the surface of Ag/TiO_2_/Ni nanopillar arrays, which provide more “hot spots” for the substrate. The XRD patterns of Ni and Ag/TiO_2_-40/Ni nanopillar arrays are shown in Figure 3, which confirms the amorphous structure of the TiO_2_ layer deposited by ALD at 250 °C.

### 3.2. Optical Properties and Interfacial Interaction Characterization

In order to investigate the optical properties of Ag/TiO_2_/Ni with different TiO_2_ thicknesses, the UV-vis spectra were carried out and shown in Figure 4. Two distinct absorption bands were found at about 304 and 485 nm. The band at 304 nm belongs to the dipole resonance of Ag, and the band at 485 nm is due to the SPR absorption of Ag corresponding to the quadrupole resonance [22,32]. The weak absorption band in the range of 665~730 nm is attributed to the absorption of Ni. Apparently, compared with other samples, the SPR absorption of Ag/TiO_2_-10/Ni exhibits an obvious redshift to 549 nm, and the absorption peak shows a broadening trend. The coupling degree and the dielectric properties of the Ag-TiO_2_ contact interface strongly depend on the thickness of the TiO_2_ layer, which plays a decisive role in electrons redistributed at the Ag-TiO_2_ interface. The obvious redshift and broadening of the absorption band are attributed to the strong coupling of Ag and TiO_2_ and the changes in electron density. Additionally, a new absorption band appeared at 423 nm on Ag/TiO_2_-10/Ni substrate, which may be attributed to the strong CT process caused by the interfacial interaction between Ag and TiO_2_; this result is similar to that of the Ag/ZnO nanoarray substrates [33].

The XPS analysis of Ag/TiO_2_/Ni substrates with different TiO_2_ thicknesses was carried out to further verify the charge distribution at the Ag-TiO_2_ interface. All peaks in the spectra were calibrated by carbon peak. Figure 5a shows the high-resolution spectra of Ag, the double peaks at 368.19 eV and 374.18 eV in the Ag 3d spectra were assigned to Ag 3d_5/2_ and Ag 3d_3/2_, respectively. The binding energy of the double peak shifted to different degrees with the introduction of TiO_2_, which indicates that the dielectric environment of the Ag-TiO_2_ interface and the electron density on the Ag surface change with TiO_2_ thickness. In Ag/TiO_2_-10/Ni substrate, the double peaks of Ag slightly shifted to the high binding energy, which is attributed to the electrons transfer from the outside nucleus to TiO_2_ and decrease the electrons density. O 1s spectra are shown in Figure 5b; the peak at 532.48 eV corresponds to the O-H bond of TiO_2_ hydroxyl groups or the oxygen chemisorption on the surface of TiO_2_ and the peak at 530.98 eV is consistent with the lattice oxygen of TiO_2_ [34]. Notably, a large number of electrons transferred from Ag to TiO_2_ increases the electron density on the TiO_2_ surface, leading to the O 1s fitting peaks of Ag/TiO_2_-10/Ni shift to the low binding energy.

### 3.3. SERS Spectra of Ag/TiO_2_/Ni Nanopillar Arrays with Different TiO_2_ Thickness

Figure 6a,b show the SERS spectra of 10^−5^ M PATP absorbed on Ag/TiO_2_-(10–0)/Ni substrates with the excitation wavelengths of 532 nm and 633 nm, and the effect of TiO_2_ thickness on SERS performance was studied. The characteristic bands of PATP molecule at 1076, 1143, 1187,1435 and 1574 cm^−1^ are clearly detected, and the band assignments are listed in Table 1 [35,36,37]. Notably, the Raman spectra measured with excitation wavelengths at 532 and 633 nm show that the SERS intensity of PATP absorbed on Ag/TiO_2_/Ni substrates varies with thickness non-monotonically and exhibits a strong TiO_2_ thickness dependence. Apparently, Ag/TiO_2_-10/Ni exhibits the strongest SERS intensity. As the TiO_2_ thickness increases, the SERS intensity shows a trend of first weakening and then increasing, and it tends to be stable at 40–60 nm. It is known that in the visible region, the strong SPR of Ag contributes largely to the electromagnetic enhancement and SERS activity. The changes in TiO_2_ thickness in the composite system significantly influence the contribution of electromagnetic enhancement and chemical enhancement to SERS, which results in the Ag/TiO_2_/Ni substrate exhibiting a TiO_2_ thickness-dependent SERS behavior.

It is worth noting that the intensity of a_1_ and b_2_ vibrational mode in Raman spectra changes significantly with TiO_2_ thickness, which is attributed to the different contributions of SPR and CT enhancements to SERS. According to the CT model proposed by Lombardi et al., only the totally symmetric vibrational modes of the molecules can be enhanced by the contribution corresponding to the Franck–Condon principle, while both totally and non-totally symmetric vibrational modes of the molecules are expected to be enhanced via Herzberg–Taller effect [11]. Compared to the Raman spectra obtained at the two excitation wavelengths, the intensity of the b_2_ vibration mode decreases significantly at 633 nm excitation, which is attributed to the weakening of CT enhancement via the Herzberg–Taller effect.

### 3.4. CT Contribution of SERS in PATP/Ag/TiO_2_/Ni System

In order to study the intensity variation in the two vibrational modes in the spectra, the Raman intensity ratio of 1143 cm^−1^ (b_2_)/1076 cm^−1^ (a_1_) vibrational bands, measured with excitation at 532 nm (red line) and 633 nm (green line), were obtained. As shown in Figure 7a, the b_2_/a_1_ ratio obtained at 532 nm excitation is significantly higher than that obtained at 633 nm excitation, which could be explained by the strong CT resonance enhancement induced at the high excitation energy between the substrate and the molecule. Moreover, the ratio changes with the TiO_2_ thickness indicate a selective enhancement in the intensity of the two vibrational modes.

The CT degree (ρCT) is used to quantitative evaluate the contribution of CT resonance enhancement on the overall SERS intensity, which was proposed by Lombardi et al., and the CT degree of vibration mode (k) can be defined by the following equation [11]:(1)$$\rho_{\mathrm{CT}}\left(k\right)=\frac{I^{k}\left(\mathrm{CT}\right)-I^{k}\left(\mathrm{SPR}\right)}{I^{k}\left(\mathrm{CT}\right)+I^{0}\left(\mathrm{SPR}\right)}$$
where k is an index for identifying individual molecular bands in the Raman spectrum, I^k^(CT) is the intensity of the Raman band, which provides additional CT resonance enhancement to the total SERS intensity. I^0^(SPR) is the intensity of a totally systematic band, and I^k^(SPR) is the intensity of the Raman band with only SPR contribution. I^k^(SPR) = I^0^(SPR) in the case of k belongs to a totally symmetric vibrational mode.

When CT resonance enhancement dominates the SERS contribution in the system, ρCT = 1 and the non-totally symmetric modes become prominent. Conversely, the totally symmetric modes dominate the spectrum far from the CT resonance and the ρCT = 0. When the SPR enhancement and CT resonance enhancement contribution equally to SERS, ρCT = 0.5.

In Ag/TiO_2_/Ni system, the a_1_ totally symmetric modes of 1143 cm^−1^ contributed by SPR independently are selected as I^0^(SPR), while the b_2_ non-totally symmetric modes of 1143 cm^−1^ are selected as the object of research on CT contribution and denoted by I^k^(SPR). The CT degree of Ag/TiO_2_/Ni composite substrates was calculated at 532 nm and 633 nm excitation wavelengths and is shown in Figure 7b. It is obvious that Ag/TiO_2_-10/Ni substrate exhibits the strongest CT contribution to SERS and the CT degree weakens to various degrees with increasing thickness. In particular, the value of ρCT exceeds 0.5 under 532 nm excitation, which suggests the CT resonance contribution in Ag/TiO_2_/Ni system has a significant effect on SERS enhancement.

### 3.5. Characterization of CT Process between Substrates and PATP Molecules

XPS analysis was studied to reveal the CT process between Ag/TiO_2_/Ni substrates and PATP molecules. O 1s and S 2p high-resolution spectra of 10^−5^ M PATP adsorbed on TiO_2_-(10, 30, 50)/Ni are shown in Figure 8a,b, and C 1s is used to calibrate the binding energy. It can be seen in Figure 8a that the characteristic peaks of O 1s spectra are located at 534.52 eV and 531.98 eV, respectively. Interestingly, compared to the O 1s spectra shown in Figure 5a, the binding energy increases significantly after adsorption of PATP molecules, which is the result of electrons transfer from substrate to PATP molecules. In addition, the peaks of Ag/TiO_2_-10/Ni shift to higher energy compared to other samples, which indicates the intense CT interaction between substrate and PATP molecule. The electrons transferred from substrate to PATP molecule, which reduces the electron density of TiO_2_, leading to the increase in O 1s binding energy.

The S 2p spectra are shown in Figure 8b; the peaks at about 161.64 eV and 162.97 eV are assigned to S 2p_3/2_ and S 2p_1/2_, which confirms that PATP successfully binds to the surface of TiO_2_ with Ti-S bonds [38]. It is notable that the binding energy of S 2p_3/2_ and S 2p_1/2_ of Ag/TiO_2_-10/Ni are slightly shifted to 162.74 eV and 161.44 eV. The reason for the shift is that interfacial CT from the substrate to PATP reduces the binding energy of the S 2p electron. The higher amount of charge transfer from Ag/TiO_2_/Ni to the adsorbed molecules, the stronger the effect of offsetting the nuclear potential and smaller binding energy of the S 2p electron [12]. Compared with other samples, the Ag/TiO_2_-10/Ni substrate exhibit the most effective charge transfer, which is consistent with the SERS result of PATP adsorbed on Ag/TiO_2_-10/Ni nanopillar arrays.

### 3.6. Charge Transfer Mechanism

In order to further elucidate the SERS mechanism of the PATP/Ag/TiO_2_/Ni system, a charge transfer model of the CT process is proposed and shown in Figure 9. With the vacuum level as a reference, the Fermi levels of Ag and Ni are −4.26 eV and −4.60 eV, respectively [39]. The conduction band (CB) and valence band (VB) of TiO_2_ are located at −4.00 eV and −7.20 eV [40,41]. The ionization potential of PATP is 7.16 eV, which is the location of the highest occupied molecular orbital (HOMO). The lowest unfilled molecular orbital (LUMO) corresponds to the strongest transition of b_2_ mode that can be borrowed is the π-π * transition at 300 nm, situating at 3.03 eV, which was demonstrated by Lombardi et al. [42]. The incident photons (532 nm, 2.33 eV and 633 nm, 1.96 eV) can easily transfer the excitation electrons of Ag and a few excited electrons of Ni through the CB of TiO_2_ to the LUMO of PATP. Another CT path that may exist in the system is that electron transfer from VB of TiO_2_ to the surface state energy levels (E_SS_) and transfer to LUMO of PATP, which is considered to increase the amounts of electrons transferred to the PATP molecule. The whole charge transfer process results in the enhanced CT resonance of the PATP/Ag/TiO_2_/Ni system, and the degree of CT resonance enhancement depends on the amounts of electrons participating in the charge transfer process in the system [43]. The photon of the 532 nm excitation wavelength possesses higher excitation energy, which can theoretically result in a stronger CT resonance enhancement. Furthermore, the change in TiO_2_ thickness leads to a significant shift of SPR and introduces a large amount of TiO_2_ E_SS_, which causes the system to exhibit different degrees of CT resonance enhancement and the TiO_2_-thickness-dependent SERS.

### 3.7. SERS Performance of Ag/TiO_2_/Ni Nanopillar Arrays

It is known that the detection capability is an indispensable property for SERS substrates, especially the high sensitivity of the substrate enables SERS to satisfy the low concentration analysis. Therefore, the SERS spectra of different concentrations of PATP molecules adsorbed on Ag/TiO_2_-10/Ni nanopillar arrays were obtained. As shown in Figure 10a, the vibrational peaks at 1076, 1143, 1390, 1435 and 1576 cm^−1^ were clearly detected at 10^−5^–10^−8^ M. The SERS signal gradually decreases with the decrease in the concentration of PATP solution, but still exhibits a relatively obvious enhancement at 10^−8^ M. Figure 10b exhibits the SERS spectra of 10^−9^–10^−11^ M PATP molecules absorbed on Ag/TiO_2_-10/Ni substrate. Although the Raman intensity decreases significantly, weak characteristic peaks can still be detected at 1002, 1184 and 1440 cm^−1^. However, these peaks can hardly be detected at 10^−11^ M. The results discussed above showcase that Ag/TiO_2_-10/Ni substrate possesses high SERS detection sensitivity and can be utilized for low concentration detection.

In addition to the high sensitivity, an excellent SERS substrate cannot be separated from the high uniformity, which ensures the reproducibility of the obtained SERS spectra and the reliability of the conclusions. The homogeneity of the Ag/TiO_2_/Ni substrates was inspected by performing 20 SERS measurements (five samples, four different positions each sample) of 10^−8^ M PATP absorbed on Ag/TiO_2_-10/Ni nanopillar arrays, as shown in Figure 11a. Figure 11b–d show the plotted intensity distribution histograms of three characteristic peaks located at 1076 cm^−1^, 1145 cm^−1^ and 1473 cm^−1^, and the relative standard deviation (RSD) values were calculated. The SERS intensity shows a high consistency, which is relatively consistent with the RSD results. The RSD values of the three characteristic peaks are 8.16%, 9.03% and 7.83%, respectively, and the average is about 8.34%. Ag/TiO_2_/Ni substrates are capable of offering excellent signal reproducibility, which is attributed to the high order and uniformity of the nanopillar arrays.

## 4. Conclusions

In this work, Ag/TiO_2_/Ni nanopillar arrays with precisely controlled TiO_2_ thickness were fabricated, and the SERS spectra of PATP absorbed on Ag/TiO_2_/Ni substrates were obtained with different excitation wavelengths. The interfacial interaction between Ag and TiO_2_ as well as the CT process of substrates and molecules were studied in PATP/Ag/TiO_2_/Ni composite system, and the TiO_2_ thickness-dependent CT mechanism was discussed. The SERS performance of Ag/TiO_2_/Ni nanopillar arrays exhibits an obvious TiO_2_ thickness dependence, which is attributed to the changes in SPR absorption and the different degrees of CT resonance enhancement caused by TiO_2_ thickness. Furthermore, Ag/TiO_2_-10/Ni composite system has the potential to be used as a SERS active substrate due to the favorable detection ability and uniformity. These results indicate that the semiconductor thickness plays an important role in the optimization of SERS performance in the composite system. This work not only provides a new method for the study of CT mechanism in composite SERS substrates but also provides a reference for the reasonable design of a composite system.

## Figures and Tables

**Figure 1 materials-15-03716-f001:**
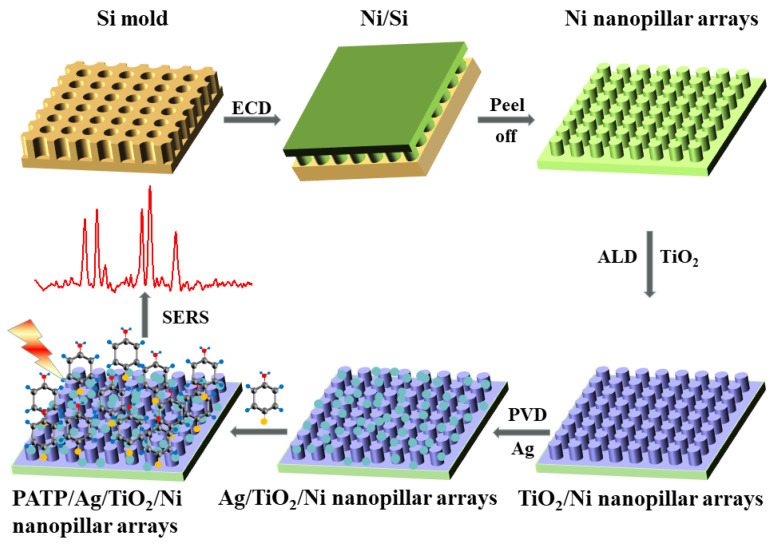
Schematic diagram of the fabrication process of Ag/TiO_2_/Ni nanopillar arrays and SERS measurement of PATP molecules.

**Figure 2 materials-15-03716-f002:**
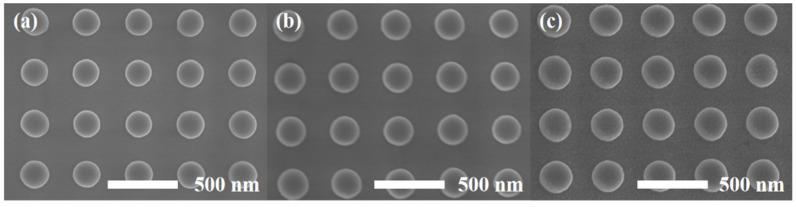
FE-SEM images of (**a**) Ni, (**b**) TiO_2_-40/Ni and (**c**) Ag/TiO_2_-40/Ni nanopillar arrays.

**Figure 3 materials-15-03716-f003:**
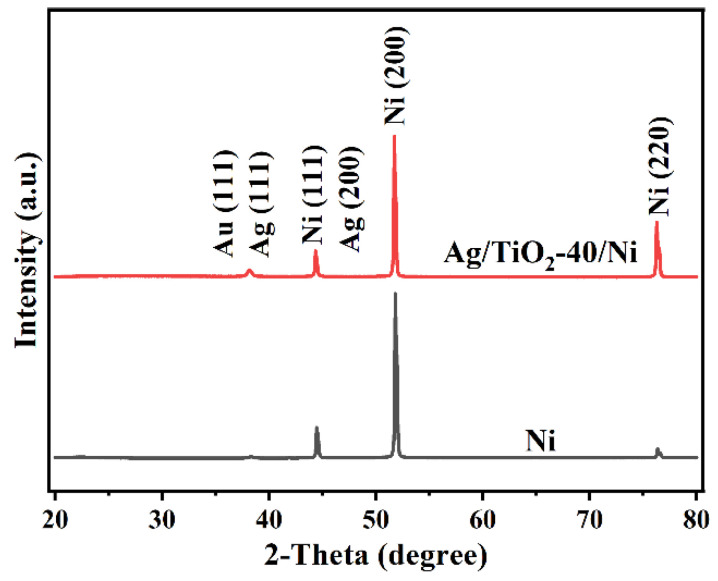
XRD patterns of Ni and Ag/TiO_2_-40/Ni nanopillar arrays.

**Figure 4 materials-15-03716-f004:**
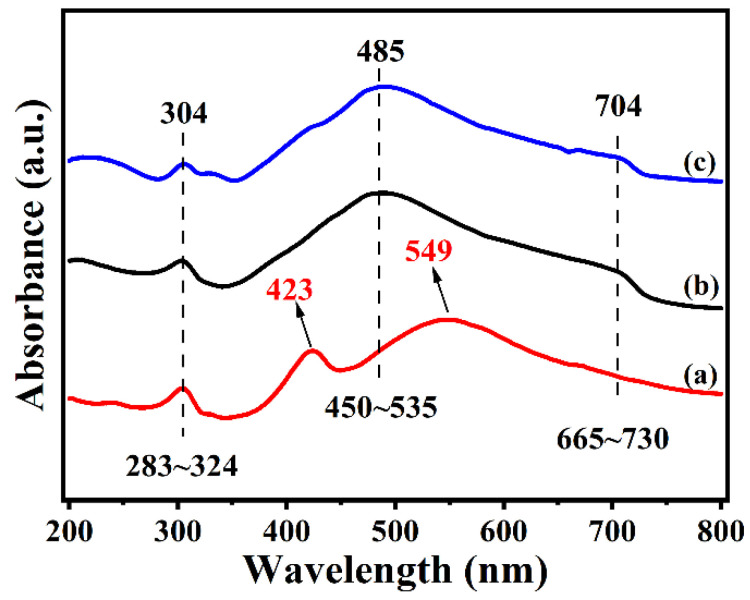
UV-vis absorption spectra of Ag/TiO_2_-10/Ni (**a**), Ag/TiO_2_-30/Ni (**b**) and Ag/TiO_2_-50/Ni (**c**).

**Figure 5 materials-15-03716-f005:**
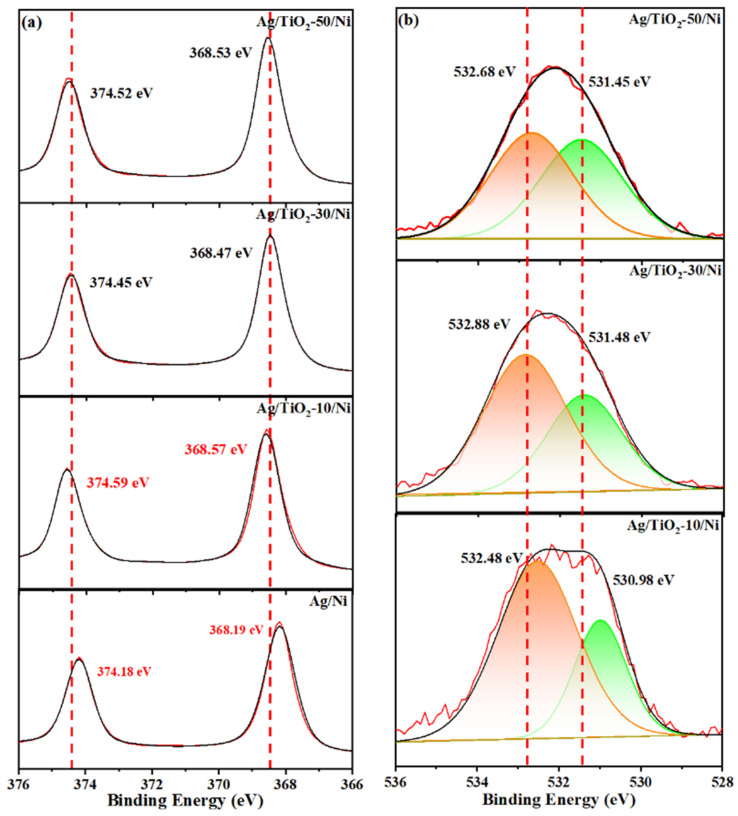
XPS spectra of Ag/TiO_2_/Ni substrates with different TiO_2_ thickness (**a**) Ag 3d, (**b**) O 1s.

**Figure 6 materials-15-03716-f006:**
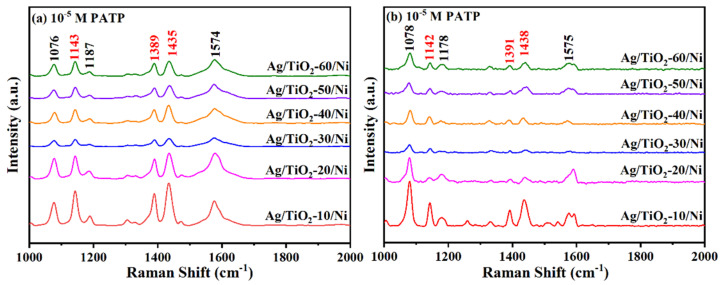
SERS spectra of 10^−5^ M PATP absorbed on Ag/TiO_2_-(10–60)/Ni nanopillar arrays with the excitation wavelengths of (**a**) 532 nm, (**b**) 633 nm.

**Figure 7 materials-15-03716-f007:**
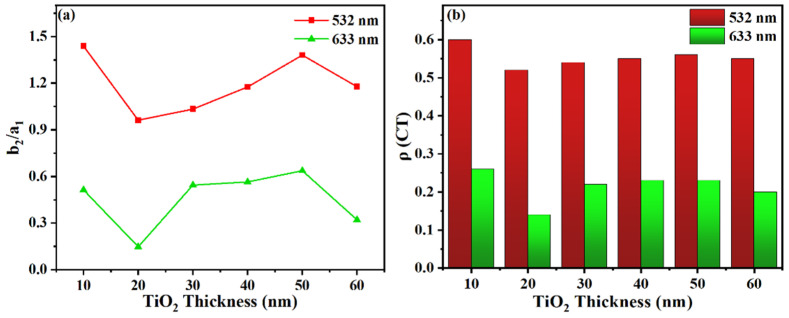
(**a**) The SERS intensity ratio of b_2_/a_1_ at 1143 cm^−1^ (b_2_ mode) and 1076 cm^−1^ (a_1_ mode) bands measured with excitation at 532 nm (red line) and 633 nm (green line); (**b**) the CT degree in Ag/TiO_2_/Ni composite substrates at those two band positions.

**Figure 8 materials-15-03716-f008:**
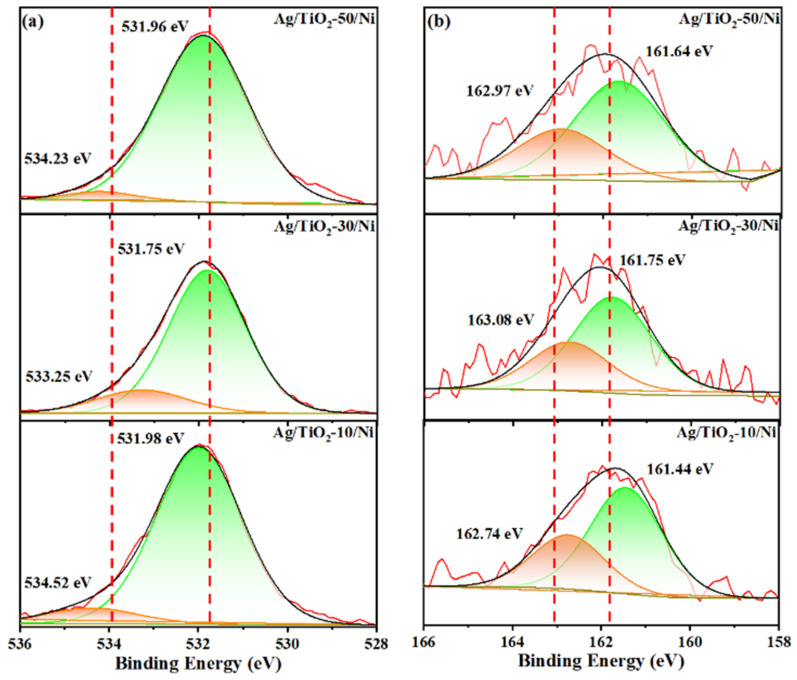
XPS spectra of 10^−5^ M PATP absorbed on Ag/TiO_2_-(10, 30, 50)/Ni nanopillar arrays (**a**) O 1s spectra, (**b**) S 2p spectra.

**Figure 9 materials-15-03716-f009:**
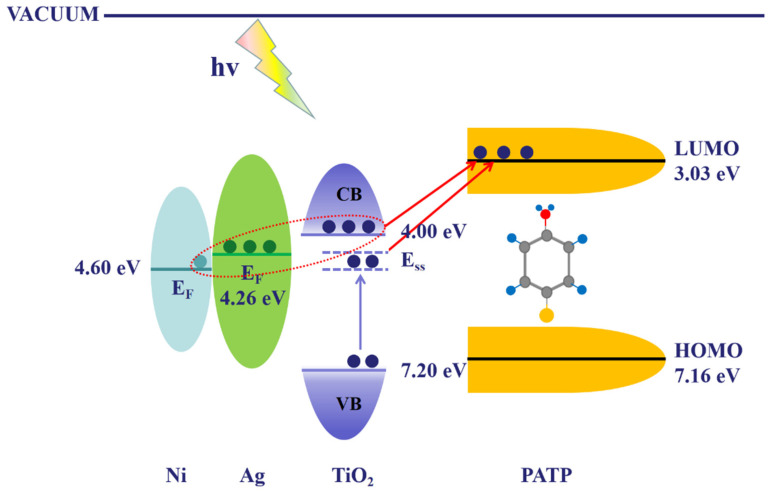
Schematic illustration of CT mechanism in PATP/Ag/TiO_2_/Ni system.

**Figure 10 materials-15-03716-f010:**
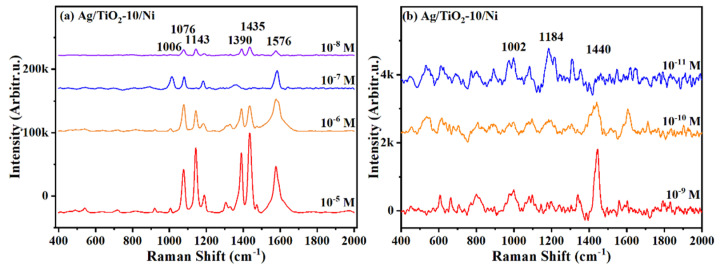
Raman spectra of PATP with various concentrations, (**a**) 10^−5^, 10^−6^, 10^−7^ and 10^−8^ M and (**b**) 10^−9^, 10^−10^ and 10^−11^ M, absorbed on Ag/TiO_2_-10/Ni nanopillar arrays.

**Figure 11 materials-15-03716-f011:**
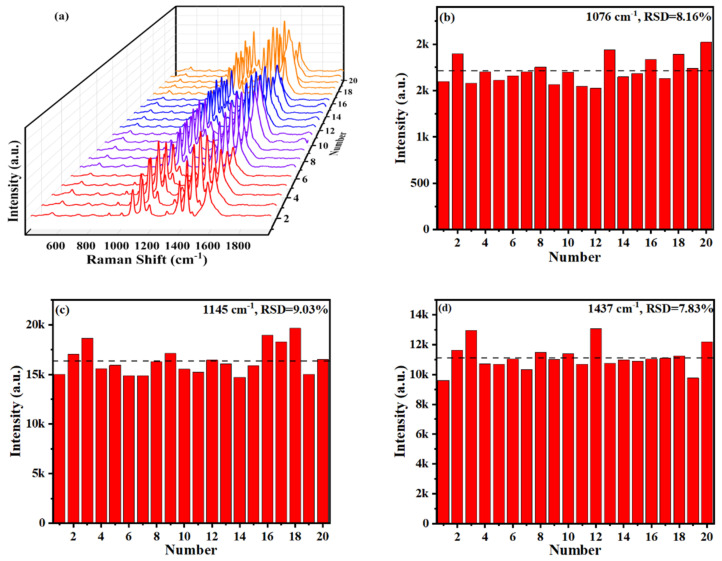
(**a**) SERS spectra of 10^−8^ M PATP molecules randomly attained at 20 positions on Ag/TiO_2_-10/Ni nanopillar arrays, SERS intensity distribution histogram, and RSD value of the characteristic bands of PATP molecules at (**b**) 1076 cm^−1^, (**c**) 1145 cm^−1^ and (**d**) 1473 cm^−1^.

**Table 1 materials-15-03716-t001:** Wavenumbers and assignments of band in a SERS spectrum of PATP [34,35,36].

Mode	Wavenumber (cm^−1^)	Band Assignments
a_1_	1006	γ (CCC) + γ (C-C)
a_1_	1088	υ (C-S)
b_2_	1147	δ (C-H)
a_1_	1176	δ (C-H)
b_2_	1395	θ (C-H) + ν (C-C)
b_2_	1438	δ (C-H) + ν (C-C)
a_1_	1592	ν (C-C)

Note: ν, stretching; δ and γ, bending; θ, rocking. For ring vibrations, the corresponding vibrational modes of benzene and the symmetry species under C^2v^ symmetry are indicated.

## Data Availability

The data presented in this study are available on request from the corresponding author.

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
