# Peer review of "TiO2 Thickness-Dependent Charge Transfer in an Ordered Ag/TiO2/Ni Nanopillar Arrays Based on Surface-Enhanced Raman Scattering"

_materials, 2022, doi:10.3390/ma15103716_

Round 1

Reviewer 1 Report

This is a fairly solid work, which should undoubtedly be recommended for publication, but after clarifying some incomprehensible points.

  1. Paragraph 3.2 and figure 3. There are inconsistencies in the position of the maxima in the figure and the data in the text. Moreover, it is difficult to explain that the 304 nm maximum is due to the band gap, because there is no corresponding increase in absorption in the region of 200 -300 nm.
  2. It is also interesting why nickel transitions in the spectrum are not considered (Fig.3). For example, Ni2+(3d8) ions from the ground state  to the excited state   have absorption at 2.5 eV in NiWO4 ( Kuzmin et al, Low Temperature Physics 42, 543 (2016); https://doi.org/10.1063/1.4959010)
  3. More information and corresponding discussion is needed about the structure of TiO2 (rutile or atanase).

Reviewer 2 Report

In this manuscript an ordered Ag/TiO2/Ni nanopillar arrays hybrid substrate was designed and the interfacial interaction between Ag and TiO2 as well as the charge transfer (CT) process of substrates and molecules were studied in PATP/Ag/TiO2/Ni composite system. The SERS performance of Ag/TiO2/Ni nanopillar arrays exhibit TiO2thickness dependence, which is attribute to the changes of SPR absorption and the different degrees of CT resonance enhancement caused by TiO2 thickness. Furthermore, Ag/TiO2-10/Ni composite system has potential to be used as a SERS active substrate due to the favorable detection ability and uniformity. As demonstrated in this manuscript, the subject is interesting and relevant to Materials. There are some important issues that should be addressed before considering this work for publication.

Here are some specific comments:

  1. The authors need to explain the novelty of their works compared to previously reported studies based on the semiconductor thickness and charge transfer contribution to SERS.
  2. Recent work related to charge transfer mechanisms in SERS assays would be helpful for this manuscript and should be cited. For example, https://doi.org/10.1021/acsami.2c02934.
  3. Why choose 4-Aminothiophenol (PATP) as a model analyte in this study? Please give an introduction.
  4. Authors mentioned that Ag/TiO2-10/Ni exhibits the strongest CT contribution to SERS and CT degree weakens to various degrees with increasing thickness. Ag/TiO2-10/Ni results are not significantly different from Ag/TiO2-50/Ni in figure 6(a) and 6(b) for both excitation wavelengths. Please explain.
  5. Less information is provided on data processing. Authors should clear that if the SERS spectra throughout the manuscript are represented without any data processing, or they are baseline corrected. It seems SERS spectra in figure5, figure 9 and figure 10 are baseline corrected, if yes, then authors should provide the information about the used baseline correction method.
  6. The symbol ϱ (Rho) is used to represent the multiple terms such as rocking vibrational mode in Table 1 and CT degree (ϱCT) throughout the section 3.4. One symbol should not be used for multiple terms representation.
  7. In line #208-209, authors mentioned that Raman spectra were measured with excitation wavelength at 514 and 633nm instead of 532 and 633nm. Please correct that.
  8. In line #267 authors mentioned that O1s and S2p high-resolution spectra of 10-5 M PATP adsorbed on TiO2-(10,30,50)/Ni are shown in Figure 6a-b. Is author referring to the figure 7a-b here?
  9. Similarly, in line #268, it can be seen in Figure 6a that the characteristic peaks of O 1s spectra are located at 532.6 eV and 530.0 eV, respectively. Is author referring to the figure 7a here?

Reviewer 3 Report

In this manuscript, an Ag/TiO2/Ni nanopillar arrays hybrid substrate was designed and fabricated. The effect of surface plasmon resonance on charge transfer mechanism as well as the charge transfer process at the substrate-molecular interface was studied by modulating the TiO2 thickness. In addition, the fabricated Ag/TiO2-10/Ni composite system has potential to be used as a SERS active substrate due to the favourable detection ability and uniformity. The authors however should consider the following suggestions before publication:

  1. The effect of Ag thickness and Ni thickness on the charge transfer should be discussed.
  2. Only the SERS performance of Ag/TiO2-10/Ni nanopillar arrays is discussed in the manuscript. The SERS performances of Ag/TiO2/Ni nanopillar arrays with different TiO2 thicknesses should be shown in the manuscript, and the difference between them should be included.
  3. The electromagnetic field distribution of the Ag/TiO2/Ni nanopillar with different TiO2 thicknesses should be included in the manuscript, which will help readers understand the effect of TiO2 thickness on the charge transfer and SERS.

Round 2

Reviewer 2 Report

The authors appropriately took all criticism into account. The manuscript is more reach in the context of TiO2- thickness-dependent charge transfer enhancement mechanism. This study would be of interest for the scientific community.

My recommendation is to accept this work with the following minor correction.

  • Include volume/issue number or DOI (https://doi.org/10.1021/acsami.2c02934) of the newly added reference#44.

Author Response

Point 1: Include volume/issue number or DOI (https://doi.org/10.1021/acsami.2c02934) of the newly added reference#44.

Response 1: Thank you for your careful check. We added DOI at the end as the reference has not yet determined the volume/ issue number. We will keep in touch with editors about format revisions in time.

The following reference have been updated in the manuscript:

Poonia, M.; Küster, T.; D. Bothun, G. Organic Anion Detection with Functionalized SERS Substrates via Coupled Electrokinetic Preconcentration, Analyte Capture, and Charge Transfer. ACS Appl. Mater. Interfaces 2022, DOI: 10.1021/acsami.2c02934.

Reviewer 3 Report

Since all my concerns are addressed, I think this manuscript can be accepted.

Author Response

Thank you for your valuable comments.